# HPLC-PDA Method for Quantification of Bioactive Compounds in Crude Extract and Fractions of *Aucklandia costus* Falc. and Cytotoxicity Studies against Cancer Cells

**DOI:** 10.3390/molecules28124815

**Published:** 2023-06-16

**Authors:** Anil Bhushan, Dixhya Rani, Misbah Tabassum, Saajan Kumar, Prem N. Gupta, Sumeet Gairola, Ajai P. Gupta, Prasoon Gupta

**Affiliations:** 1Natural Products and Medicinal Chemistry Division, CSIR-Indian Institute of Integrative Medicine, Canal Road, Jammu 180001, India; 2Academy of Scientific and Innovative Research (AcSIR), Ghaziabad 201002, India; 3Pharmacology Division, CSIR-Indian Institute of Integrative Medicine, Canal Road, Jammu 180001, India; 4Drug Testing Laboratory, CSIR-Indian Institute of Integrative Medicine, Jammu 180001, India; 5Plant Science and Agrotechnology Division, CSIR-Indian Institute of Integrative Medicine, Canal Road, Jammu 180001, India

**Keywords:** *Aucklandia costus*, RP-HPLC, quantification, linearity, dehydrocostus lactone, costunolide, syringin, 5-hydroxymethyl-2-furaldehyde, SRB assay

## Abstract

*Aucklandia costus* Falc. (Synonym: Saussurea costus (Falc.) Lipsch.) is a perennial herb of the family Asteraceae. The dried rhizome is an essential herb in the traditional systems of medicine in India, China and Tibet. The important pharmacological activities reported for *Aucklandia costus* are anticancer, hepatoprotective, antiulcer, antimicrobial, antiparasitic, antioxidant, anti-inflammatory and anti-fatigue activities. The objective of this study was the isolation and quantification of four marker compounds in the crude extract and different fractions of *A. costus* and the evaluation of the anticancer activity of the crude extract and its different fractions. The four marker compounds isolated from *A. costus* include dehydrocostus lactone, costunolide, syringin and 5-hydroxymethyl-2-furaldehyde. These four compounds were used as standard compounds for quantification. The chromatographic data showed good resolution and excellent linearity (r^2^ ˃ 0.993). The validation parameters, such as inter- and intraday precision (RSD < 1.96%) and analyte recovery (97.52–110.20%; RSD < 2.00%),revealed the high sensitivity and reliability of the developed HPLC method. The compounds dehydrocostus lactone and costunolide were concentrated in the hexane fraction (222.08 and 65.07 µg/mg, respectively) and chloroform fraction (99.02 and 30.21 µg/mg, respectively), while the *n*-butanol fraction is a rich source of syringin (37.91 µg/mg) and 5-hydroxymethyl-2-furaldehyde (7.94 µg/mg). Further, the SRB assay was performed for the evaluation of anticancer activity using lung, colon, breast and prostate cancer cell lines. The hexane and chloroform fractions show excellent IC_50_ values of 3.37 ± 0.14 and 7.527 ± 0.18 µg/mL, respectively, against the prostate cancer cell line (PC-3).

## 1. Introduction

Plants serve as a source of novel bioactive metabolites. These bioactive molecules can serve either directly as leads or as active pharmacophore templates, which have immense potential for the treatment of various human diseases, especially cancer and infectious diseases [1,2]. The advent of new techniques and tools has revolutionized the screening of natural products [3]. Drug discovery from natural products involves the biological screening of a library of extracts to identify a biologically active hit, which is further fractionated to enrich the active constituents [4]. At present, only 6% of plant species have been assessed for their biological activities [5,6]. This clearly indicates that there is a huge unearthed potential for the discovery of bioactive chemicals from plants to be converted into druggable entities [7].

Natural product chemists target the enriched fractions of bioactive extracts and further isolate the pure substances responsible for biological activity [8]. Secondary metabolites are accumulated within different parts of plants, so the identification of metabolite-rich parts is essential for the isolation of bioactive leads [9]. Natural products and their semisynthetic derivatives have played a major role in drug discovery [10,11]. Drugs that are truly synthetic in origin are only 25% of available drugs, while the rest of the drugs are either from natural sources or natural-product-inspired drugs [2]. Various ailments such as diabetes [12], neurological diseases [13], genetic diseases [14], cardiovascular diseases [15], metabolic disorders [16], immunological diseases [17], inflammation and related diseases [18], oncological diseases [19], viral diseases, bacterial and fungal infections, etc. [20], are important areas in which natural-product-based drugs are profusely used for treatment.

Cancer is a global problem that is increasing at an alarming rate in both undeveloped and developed countries. The WHO estimates that by 2040, the rate of incidence may reach up to 29.5 million new cases every year, leading to 16.5 million deaths, if the dearth of potent therapeutics prevails [21]. However, the chemotherapeutic agents for cancer treatment have serious side effects leading to hair loss, neuropathy, dry eye, dizziness, headache, cardiac toxicity, neutropenia, gastrointestinal lesions, etc. [22]. Medicinal plants are an important source of anticancer therapeutics. The difficulties associated with natural product development include the decline in biodiversity and the unavailability of standard procedures for the detection and isolation of pure bioactive compounds from their natural sources. It is evident that 65% of FDA-approved anticancer leads that are currently on the market originate only from natural sources [23,24]. A variety of phytochemicals, such as alkaloids, terpenoids, taxanes, phenolic compounds and glucosinolates, extracted from various plant species have been found to be effective in cancer treatment [25,26].

The genus Saussurea contains around 400 species that are distributed worldwide. At least 300 species are found only in India, China and the Tibet region [27]. Many species of the genus Saussurea are used in traditional medicines due to their biological properties [28]. *A. costus*, commonly known as ‘kuth’, is an important medicinal species of this genus. On the Indian subcontinent, *A. costus* is distributed in parts of the western Himalayas and grows at an altitude ranging from 2600 to 4000 m [29]. It is found abundantly in the Kashmir region and is commercially cultivated in Uttarakhand, Himachal Pradesh and Sikkim. *A. costus* was observed extinction threats due to overexploitation for trade [30]. The plant is a tall perennial herb reaching heights of up to 1 to 2.2 m and has dark-purple-colored flowers arranged in terminal and axillary heads. The dried roots of *A. costus* are known as Mu Xiang in the Traditional Chinese system of medicine [31]. The plant *A. costus* has been extensively used in different traditional systems of medicine since ancient times, so numerous pharmacological activities of this plant are found in the literature data [32]. *A. costus* rhizomes have been widely used in the treatment of various ailments, such as leprosy, cough and cold, malaria, persistent hiccups, stomachache, toothache, typhoid fever, chest congestion, etc. [33]. In addition, it is used as an antispasmodic in asthmatic patients and also in the treatment of cholera, gout, erysipelas, etc. [34]. The compounds isolated from *A. costus* have been found to be effective against a wide range of cancers, such as ovarian, pancreatic, prostatic, colon and bladder cancer, leukemia, etc. [35]. Other important reported biological activities of *A. costus* include antimicrobial, antiulcer, hepatoprotective and cardioprotective effects, among others [36]. Sesquiterpenes are recognized as important markers in species belonging to the Asteraceae family [37]. These sesquiterpenes have been the focus of drug discovery because of their structural diversity (e.g., germacrane, eudesmane and guaiane skeletons) [38]. The chemical compounds isolated and identified from *A. costus* include sesquiterpenes (alantolactone, dehydrocostus lactone, costunolide, cynaropicrin, saussurealdehyde, etc.), flavonoids (kaempferol, rutin, quercetin, luteolin, etc.), phytosterols (stigmasterol, lappasterol, β-sitosterol, pregnenolone and lappasterol((24*S*)-Stigmasta-5,9(11)-dien-20,28-endo-3β-ol)), along with other constituents, such as lignans and phenolics [39]. 

For the utilization of herbal drugs, it is essential to generate a chemical profile using marker compounds. The plant metabolite matrix is very complex, and its composition can be conveniently analyzed by using an HPLC instrument [40]. This study aimed to develop an HPLC-PDA analytical method for the quantification of the crude extract and fractions based on isolated marker compounds and the evaluation of the cytotoxic potential of the crude extract and fractions of a widely used medicinal herb, *A. costus*, on various cancer cell lines. A new, simple and reliable reverse-phase HPLC-PDA method was developed for the quantification of marker compounds in the crude extract and its fractions. The marker compounds were isolated by purifying the concentrated fractions of the crude extract. The isolated compounds were dehydrocostus lactone, costunolide, syringin and 5-hydroxymethyl-2-furaldehyde. Compound characterization was performed by using modern spectroscopic techniques, such as 1D and 2D NMR spectroscopy, to generate molecular structures, along with HRMS data, which confirmed the molecular formulas of the isolated compounds. Further, the purity profiles of the isolated compounds were also assessed prior to their quantification. 

The quantification of marker compounds in the crude extract and fractions was performed by using a new, simple and reliable reverse-phase HPLC-PDA method. The developed method was found to be suitable for the quantification of the marker compounds dehydrocostus lactone, costunolide, syringin and 5-hydroxymethyl-2-furaldehyde in the crude extract and fractions of *A. costus*. Cytotoxicity studies were performed on lung cancer (A549), colon cancer (HCT-116), breast cancer (MDA-MB-231) and prostate cancer (PC-3) cells using the SRB assay. The cytotoxicity results revealed the high anticancer potential of the chloroform and hexane fractions, while the butanol fraction did not possess significant cytotoxicity. There are no reports available on the use of a reverse-phase high-performance liquid chromatography method for the simultaneous analysis and quantification of the above-mentioned four marker compounds in the *A. costus* crude extract and its fractions.

## 2. Results

### 2.1. Structure Elucidation of Marker Compounds Isolated from A. costus

The fractions obtained from the crude extract upon purification yielded secondary metabolites, including dehydrocostus lactone (compound 1), costunolide (compound 2), syringin (compound 3) and 5-hydroxymethyl-2-furaldehyde (compound 4). The chemical structures of these marker compounds were elucidated using detailed spectroscopic studies, including 1D and 2D NMR, along with HRMS experiments.

Compound 1 was isolated as a yellow gummy substance. Its molecular formula C_15_H_18_O_2_ was deduced by positive-mode HRMS (*m/z* 231.30) [M+H]^+^ (calcd. for C_15_H_19_O_2_^+^, 231.20), in combination with ^1^H and ^13^CNMR (DEPT) spectral data (Appendix A). The 15 C-atoms were identified as 7 CH_2_ (3 olefinic) and 4 CH groups, as well as 4 quaternary carbon atoms (3 pertaining to olefinic carbon and 1 CO group). These quaternary C-atoms at (C) 145.54, 151.2 and 149.2 ppm and the signal for olefinic secondary carbon atoms at 109.5, 120.1 and 112.6 ppm indicated the presence of three exocyclic double bonds. The final assignments of hydrogen atoms to their corresponding carbon atoms were assigned using HSQC correlations, and neighboring proton systems were confirmed by COSY correlations. While HMBC correlations confirmed the overall skeleton, further NOESY correlations were used to determine the stereochemical orientation of compound 1. The aforementioned spectral data suggest that compound 1 is a sesquiterpene (guaiane type). Compound 1 was further identified as dehydrocostus lactone by a comprehensive spectral analysis and a comparison with the literature data [41,42]. The purity of compound 1 was found to be greater than 94% (Appendix A).

^1^H NMR (400 MHz, CDCl_3_): δH 6.22 (1H, d, *J* = 3.5 Hz, H-13a), 5.49 (1H, d, *J* = 3.2 Hz, 13b), 5.26 (1H, d, *J* = 1.8 Hz, H-15a), 5.07 (1H, d, *J* = 1.4Hz, H-15b), 4.91 (1H, brs, H-14a), 4.81 (1H, brs, 14b), 3.96 (1H, t, *J* = 9.2 Hz, H-5), 2.95–2.85 (3H, m, H-4, 6, 8a), 2.53–2.49 (2H, m, H-2a, 8b), 2.29–2.12 (2H, m, H-2a, 10), 1.96–1.87 (2H, m, H-7a, 7b), 1.47–1.39 (1H, m, H-1b), 1.27–1.20 (1H, m, H-1a); ^13^C NMR (100 MHz, CDCl_3_): δC 170.3 (C-12), 151.2 (C-11), 149.2 (C-3), 139.8 (C-9), 120.2 (C-13), 112.6 (C-15), 109.5 (C-14), 85.3 (C-5), 52.1 (C-4), 47.6 (C-6), 45.2 (C-10), 36.2 (C-2), 32.7 (C-8), 30.9 (C-7), 30.3 (C-1).

Compound 2 was obtained as a white amorphous solid. A molecular formula of C_15_H_20_O_2_ was deduced by positive-mode HRMS (*m/z* 233.30 [M + H]^+^ (calcd for C_15_H_21_O_2_^+^, 233.1550)), in combination with ^1^H and ^13^CNMR (DEPT) experiments (Appendix A). The analysis of the NMR data revealed that compound 2 has a close structural resemblance to compound 1. A noticeable difference between compound 2 and compound 1 is reflected by the presence of two additional singlet methyl groups. The presence of quaternary carbon at 140.16 and the signal for an olefinic secondary C-atom at 119.55 (C13) indicate the presence of an exocyclic carbon–carbon double bond in compound 2. The final structure of compound 2 was deduced by using the 2D NMR technique. Compound 2 was further identified as a germacrane type of sesquiterpene. A closer examination of available literature reports confirmed isolated compound 2 as costunolide [41,42]. The purity of isolated costunolide was found to be greater than 93% (Appendix A).

^1^H NMR (400 MHz, CDCl_3_): δ H 6.25 (1H, d, *J* = 3.6 Hz, H-13a), 5.53 (1H, d, *J* = 3.2 Hz, 13b), 4.85 (1H, dd, *J* = 11.2, 4.1 Hz, H-1), 4.75 (IH, d, *J* = 9.9 Hz, H-5), 4.59 (1H, t, *J* = 9.3 Hz, H-6), 2.58 (1H, tt, *J* = 3.1, 2.1 Hz, H-7), 2.46 (1H, dd, *J* = 13.3, 6.0 Hz, H-9), 2.34–2.03 (7H, m, H-2, 3, 8, 9d), 1.71 (3H, d, *J* = 1.1Hz, H-15), 1.42 (3H, s, H-14); ^13^C NMR (100 MHz, CDCl_3_): δ C 170.5 (C-12), 141.4 (C-11), 140.3 (C-4), 136.9 (C-10), 127.3 (C-5), 127.2 (C-1), 119.5 (C-13), 81.9 (C-6), 50.49 (C-7), 41.1 (C-9), 39.5 (C-3), 28.2 (C-8), 26.2 (C-2), 17.3 (C-15), 16.1 (C-14).

Compound 3 was isolated as an amorphous solid from the *A. costusn*-butanol fraction. Its molecular formula C_17_H_23_O_9_ was deduced by positive-mode HRMS (*m/z* 395.1323) [M+Na]^+^ (calcd. C_17_H_23_O_9_Na^+^, 395.1323), in combination with ^1^H and ^13^CNMR (DEPT) spectral data (Appendix A). The signals of a glucopyranosyl (Glc) moiety were observed in the ^1^H and ^13^CNMR spectra. The ^1^H NMR spectrum showed a typical singlet signal that integrates two protons appearing at δH 6.75 ppm (H-5, H-9), which indicates the presence of a symmetric tetra-substituted aromatic ring, and another singlet at δH 3.85 ppm (6H) is attributable to two methoxy substituents. The observation of two characteristic doublets of triplets at δH 6.54 ppm (1H, *J* 1 = 15.9 Hz) and δH 6.32 ppm (1H, *J* 1 = 15.9 Hz and *J* 2 = 5.6 Hz, H-2) suggests the presence of a trans-disubstituted double bond directly linked to a methylene group. The ^13^C NMR spectrum obtained in methanol showed the characteristic peak of anomeric carbon C1′ at δc 105.45 ppm and five peaks appearing between δc 62.71 ppm and δc 78.51 ppm attributable to C2′, C3′, C4′ and C5′ of a glucopyranose moiety, indicating that compound 3 is an aromatic glucoside derivative. The comparison of the NMR data of compound 3 with that in the literature confirmed it as syringin [43]. The purity profile of isolated compound was found to be greater than 98% (Appendix A).

^1^H NMR (400 MHz, CD_3_OD): δ H 6.75 (2H, s, H-2′,6′), 6.54 (1H, d, *J* = 16.0 Hz, H-3), 6.32 (1H, m, H-2), 4.85 (1H, s, H-1′′), 4.21 (2H, dd, *J* = 5.6, 1.2 Hz, H-1), 3.85 (6H, s, 3′, 5′-OCH_3_), 3.77 (1H, dd, *J* = 12.0, 2.4 Hz, H-6′′), 3.65 (1H, dd, *J* = 12.0, 5.2 Hz, H-6′′), 3.46–3.39 (4H, m, H-2′′to H-5′′); ^13^C NMR (100MHz, CD_3_OD): δ C 154.5 (C-3′,5′), 135.9 (C-1′), 135.3 (C-4′), 131.4 (C-3), 130.1(C-2), 105.5 (C-2′,6′), 105.4 (C-1′′), 78.5 (C-5′′), 77.9 (C-3′′), 75.8 (C-2′′), 71.4 (C-4′′), 63.7 (C-6′′), 62.7 (C-1), 57.1 (3′, 5′-OCH_3_);

Compound 4 was obtained as a dark greenish-brown liquid from the *n*-butanol fraction. The molecular formula C_6_H_6_O_3_ was deduced by HRMS m/z 127.0407 [M+H]^+^ (calcd for C_6_H_7_O_3_^+^, 127.0407) in positive-mode HRMS and its ^1^H and ^13^CNMR (DEPT) spectral data (Appendix A). The ^1^H NMR spectrum revealed the presence of two doublets observed at 7.22 and 6.60 ppm, characteristic of an aromatic region, with a coupling constant of 3.52 Hz, while a singlet at 9.54 ppm confirms the presence of an aldehydic functionalityin the isolated molecule. A doublet appears at 4.50 ppm with a coupling constant of 5.18 Hz, and a triplet occurs at 5.60 ppm with a coupling constant of 5.78 Hz. The ^13^C and DEPT spectra confirmed the presence of two quaternary carbon atoms, one methylene and three methines, in the isolated molecules. The above-given NMR data arecharacteristic of a furan skeleton containing hydroxymethyl and methyl aldehyde substitutions. The detailed spectral analysis of compound 4 and a comparison with literature reports confirmed it as 5-hydroxymethyl-2-furaldehyde [44]. The purity of compounds 3 and 4 was found to be more than 98% (Appendix A). The chemical structures of the four isolated standard compounds are shown in Figure 1.

^1^H NMR (DMSO d_6_, 400 MHz): δ 9.54 (1H, s, H-1), 7.38 (1H, d, *J* = 3.2 Hz, H-3), 6.58 (1H, d, *J* = 3.6 Hz, H-4), 4.61 (4H, s, H-6); ^13^C NMR (DMSO d_6_, 100 MHz): δ 179.5 (C-1), 163.3 (C-5), 154.1 (C-2), 124.8 (C-3), 111.2 (C-4), 57.7 (C-6).

### 2.2. Identification of Marker Compounds in Crude Extract

The HPLC chromatogram for marker compounds revealed a retention time of 23.55 min for dehydrocostus lactone, while for costunolide, the retention time was found to be 22.84 min. For syringin and 5-hydroxymethyl-2-furaldehyde, the peak retention times were found to be 10.85 min and 9.56 min, respectively, as shown in Figure 2. The optimized chromatographic conditions showed well-separated peaks of standard compounds present in the crude extract (Figure 3) and fractions (Appendix A), which can easily be identified on the basis of the retention times and UV spectra of standard compounds. Arun time of 35 min was found to be effective for both qualitative and quantitative analyses. The mobile phase gradient conditions were 0–3 min 100% A, 3–9 min 40% A, 9–14 min 40% A, 14–20 min 20% A, 20–28 min 0% A, 28–31 min 0% A, 31–33 min 100% A and 33–35 min 100% A. Formic acid was found to be a suitable buffering agent to remove the peak tailing, and a good baseline resolution was obtained. The retention time of dehydrocostus lactone was found to be 23.55 min, and for costunolide, the retention time was observed to be 22.84 min. For syringin and 5-hydroxymethyl-2-furaldehyde, the peak retention times were found to be 10.85 min and 9.56 min, respectively.

### 2.3. Correlation Coefficient (r^2^), Limit of Detection (LOD) and Limit of Quantification (LOQ)

The calibration curve was constructed by plotting the peak area versus the concentration of standard compounds. The correlation coefficient (r^2^) was calculated and found to be consistently greater than 0.993, showing good linearity over a range of 0.98–500 µg/mL (Appendix A). The above-mentioned parameters were obtained by using the standard deviation of the calibration curve intercept values (Sn). The equation for the calculation is as follows: LOD = 3 Sn/u, LOQ = 10 Sn/u, where u is the slope of the calibration curve. The results are given below in Table 1.

### 2.4. Quantification of Marker Compounds in Crude Extract and Fractions

The amounts of marker compounds in the ethanolic crude extract and all fractions of *A. costus* were determined (Figure 4). For the crude extract, dehydrocostus lactone was found to bethe major metabolite 15.27 ± 0.12 µg/mg, while costunolide has a content of 4.33 ± 0.07 µg/mg. The other two marker compounds, syringin and 5-hydroxymethyl-2 furaldehyde, were assessed to be 6.39 ± 0.03 µg/mg and 2.13 ± 0.01 µg/mg, respectively. The relative contents of marker compounds in different fractions were also determined, and it was found that dehydrocostus lactone and costunolide were abundantly present in hexane and chloroform fractions, while syringin and 5-hydroxymethyl-2-furaldehyde were extensively distributed in the butanol fraction.

#### System Suitability

System suitability was assessed to confirm that the HPLC instrumentation used and the applied method were capable of generating data within the permissible limits of the experiment. Accordingly, the observed parameters, i.e., asymmetry, signal-to-noise ratio and theoretical plate count, were found to be within acceptable limits. The results are summarized in Table 2.

### 2.5. Intra- and Interday Precision

The intraday precision measurements for dehydrocostus lactone at 10.00, 100.00 and 250 µg/mL, costunolide at 5.00, 50.00 and 125.00 µg/mL, syringin at 10.00, 100.00 µg/mL and 250.00 µg/mL and 5-hydroxymethyl-2-furaldehyde at 5.00, 50.00 and 125.00 µg/mL have relative standard deviations (%RSD) ranging from 0.05 to 1.70%. Further, the interday precision values of dehydrocostus lactone at 10.00, 100.00 and 250.00 µg/mL, costunolide at 5.00, 50.00 and 125.00 µg/mL, syringin at 10.00, 100.00 and 250.00 µg/mL and 5-hydroxymethyl-2-furaldehyde at 5.00, 50.00 and 125.00 µg/mL have relative standard deviations (%RSD) ranging from 0.05 to 1.85%. The percentage RSD values of intra- and interday precision were found to be less than 2%, which reflects the good precision and accuracy of the instrument used for the analysis. The results are given in Table 3.

### 2.6. Analyte Recoveries

The recoveries of analytes were determined by spiking the original concentrations at three different concentration levels. The recoveries of dehydrocostus lactone at 0.50, 0.85 and 1.130 µg/mL solution were 110.0%, 108.96% and 104.22%%, and those of costunolide at 0.85, 1.50 and 2.00 µg/mL solution were 97.62%, 105.02% and 102.38%. The syringin recovery percentages for 0.45, 0.90 and 1.35 µg/mL solution were 101.77%, 103.03% and 102.43%, and 5-hydroxymethyl-2-furaldehyde recoveries for 0.50, 1.25, and 1.75 µg/mL solution were found to be 110.20%, 97.52% and 101.14%. The percentage RSD values for the recovery of the analytes range from 0.37 to 1.94 (Table 4).

### 2.7. Specificity

The comparison of HPLC chromatograms of pure standard compounds and that of crude extract and bioactive fractions confirmed that there is no interference of the matrix constituents of the extract with the peaks of the pure compounds. The retention times of marker compounds in the crude extract were found to be the same as those of the pure compounds, which established the specificity of the developed HPLC method.

### 2.8. Stability of Analyte Solutions

The analyte stabilities were assessed by obtaining the percentage differences in peak areas under storage conditions of room temperature and 4 °C for time intervals of 0, 6, 12, 24 and 48 h. The percentage RSD values observed for peak areas after 48 h for dehydrocostus lactone were 1.84% at rt and 1.95% at 4 °C, and those for costunolide were 2.22% at rt and 2.52% at 4 °C. The %RSD values of peak areas for syringin were 2.24% at rt and 3.78% at 4 °C, and those for 5-hydroxymethyl-2-furaldehyde were 2.69% at rt and 2.73% at 4 °C. Since the peak areas decreased overtime, it is recommended that samples be stored at 4 °C and used within 12 h to obtain good precision in data analysis (Table 5).

### 2.9. Anticancer Activity of A. costus Crude Extract and Fractions

The anticancer activity of the crude extract and fractions were evaluated using the Sulforhodamine B test [45]. The hexane fraction exhibited significant cytotoxicity against HCT-116, MDA-MB-231 and PC-3 cell lines, while the chloroform fraction showed significant anticancer potential against the PC-3 cell line. However, the butanol fraction did not exhibit any significant anticancer activity. The dose vs. response curves for the calculation of IC_50_ are given in Figure 5. This is the first report of the anticancer activity of the different fractions and crude extract of *A. costus* quantified for the four marker compounds dehydrocostus lactone, costunolide, syringin and 5-hydroxymethyl-2-furaldehyde. The results are summarized in Table 6 below.

## 3. Discussion

The chromatographic method developed for simultaneous analysis was validated for specificity, linearity (from the calibration curve), precision, accuracy, the recovery of the analytes and the stability of the solution according to guidelines of the International Conference on Harmonisation (ICH) [46,47]. The specificity was examined by comparing the retention times and analyzing the absorption spectra of the chromatograms obtained for the samples and the standard solution in the range from 190 to 440 nm. The linearity was confirmed by obtaining the correlation coefficient (r^2^) of the calibration curve of the peak area and standard concentrations. The linearity was analyzed in a concentration range of 0.98–500 µg/mL. The samples were injected 10 times. The calibration curves so obtained were analyzed by applying the regression equation y = mx + c, where c is the intercept of the regression line, y is the peak area, m is the slope and x is the sample concentration. The calibration plots of the four standard compounds have correlation coefficients (r^2^) greater than 0.99, which established good linearity [48].

The sensitivity of the method was evaluated in terms of limits of detection (LODs) and limits of quantification (LOQs) [49]. The equations for their calculation areas follows: LOD = 3 Sn/u and LOQ = 10 Sn/u, where u is the slope of the calibration curve, and Sn is the standard deviation of the y-intercept [50]. Precision data were calculated for both inter- and intraday analysis. The intraday precision test was carried out by analyzing three different concentrations (Low, Medium and High) of each compound injected thrice on the same day, while interday precision was obtained by injecting the same three concentrations in triplicate on three consecutive days [50,51]. Recovery studies of the analytes were performed so as to confirm their percentage recoveries [52]. The equation used for the calculation of analyte recovery is as follows:(Analyte amount observed in spiked sample − analyte amount in original sample)/spiked amount of analyte × 100(1)

Method development is essential in chromatographic separation, and finally, the quantification of compounds from *A. costus* was achieved. The analytical conditions were optimized by using a large number of methods. Obtaining good sensitivity detection in the region of 205–210 nm is desirable. The HPLC-PDA method has advantages such as reproducibility, low cost and reliability [53,54]. A variety of solvents, column temperatures and run times were applied so as to obtain the simultaneous separation of analytes present in the crude extract. A variety of solvent system combinations using methanol/water and acetonitrile/water containing different buffering agents, such as trifluoroacetic acid, formic acid and acetic acid, were used. Arun time of 35 min was found to be satisfactory for the elution of the compounds. Different flow rates (0.5, 0.8 and 1 mL/min) and column temperatures (25, 30, 35 and 40 °C) were assessed so as to obtain well-resolved spectra in the shortest possible time period of sample injection. Elution was performed using a mobile phase of 0.1% formic acid in water (A) and methanol. Formic acid was found to be a suitable buffering agent to remove the peak tailing [55], and a good baseline resolution was obtained.

The isolated compounds dehydrocostus lactone, costunolide, syringin and 5-hydroxy-2-methylfuraldehyde are reported to have a wide range of pharmacological actions. The activities reported for dehydrocostus lactone and costunolide include antiviral [56], anticancer [57], anti-inflammatory [58] and antiulcer effects [59]. Another important marker compound, syringin, is a phenylpropanoid glycoside also reported in various medicinal plants, such as *Edgeworthiachrysantha, Acanthopanax senticosus*, *Musa paradisiaca*, etc. Important biological properties reported for syringin are anti-inflammatory and antidiabetic effects [60]. 5-Hydroxy-2-methylfuraldehyde (5-HMF) is a marker compound isolated from the butanol fraction of *A. costus*. It is one of the constituents of natural medicinal honey. 5-HMF is a furan compound and can be derivatized for the production of new therapeutic agents for controlling the spread of cancer [61,62]. Another important biological activity of 5-HMF is its potential to increase the Hemoglobin affinity of sickle cells [63].

The study of the crude extract and fractions is essential, as the synergistic effect of several constituents enhances the overall therapeutic value as compared to individual compounds [64]. This study evaluated the cytotoxic potential (expressed as the IC_50_ value)of the crude ethanolic extract and different fractions (rich in isolated marker compounds) of the rhizome of *A. costus*. The results reveal that hexane and chloroform fractions have the highest anticancer activity, with IC_50_ values of 3.37 and 7.527 µg/mL, respectively, against the prostate cancer cell line, whereas the crude ethanolic extract has an IC_50_ value of 52.99 µg/mL against the prostate cancer cell line. The standard drug paclitaxel used in this assay has an IC_50_ value of 0.005 µg/mL.

The crude extract and obtained fractions were also tested on lung, colon and breast cancer cell lines. The results reveal that the hexane fraction has IC_50_ values of 20.09, 4.71 and 11.87 µg/mL against lung, colon and breast cancer cell lines, respectively. The chloroform fraction has IC_50_ values of 11.87, 11.53 and 12.17 µg/mL, respectively, against lung, colon and breast cancer cell lines. The butanol fraction does not show significant cytotoxicity against any of the cell lines (IC_50_ > 100 µg/mL). The crude ethanolic extract has IC_50_ values of 64.05, 52.99 and 35.76 µg/mL, respectively, against colon, prostate and breast cancer cell lines, whereas it has no significant cytotoxicity against the lung cancer cell line. The standard drug paclitaxel has IC_50_ values of 0.01, 0.056 and 0.049 µg/mL against lung, colon and breast cancer cell lines, respectively. The butanol fraction has no significant cytotoxicity against any of the cancer cell lines.

Thus, it is hypothesized that the cytotoxicity of the crude extract and fractions is directly related to the content of sesquiterpenes. The results confirm the medicinal potential of *A. costus* in the treatment of cancer, as reported in the literature [35,65]. Hence, *A. costus* can provide leads for the generation of new anticancer compounds.

## 4. Materials and Methods

### 4.1. Plant Material

The rhizomes of *Aucklandia costus* Falc. were collected from the Kishtwar district of Jammu and Kashmir on 22 December 2018. The plant material was authenticated by Dr. Sumeet Gairola, Plant Sciences and Agrotechnology Division, CSIR-IIIM, Jammu, and voucher specimens (No. RRLH-24022) were deposited at the internationally recognized Janaki Ammal Herbarium (RRLH) at CSIR-IIIM, Jammu.

### 4.2. Chemicals, Reagents and Standard Compounds

For the HPLC analysis, solvents such as water, methanol, acetonitrile, trifluoroacetic acid, formic acid, acetic acid and isopropanol were HPLC-grade and purchased from Merck (Sigma Aldrich, Benguluru, India). The overall purity profiles of the standard compounds used for quantification were found to be greater than 93%.

### 4.3. NMR and HRMS Instrumentation

High-resolution mass spectra were obtained on an Agilent 6540 (Q-TOF) mass spectrometer in the electrospray (ESIMS) mode. All analytes were assessed by thin-layer chromatography (TLC) on silica gel 60 F_254_ (0.25 mm thick, Merck) with spots visualized by UV 254 and 366 nm, and anisaldehyde reagent was used as the development reagent. Column chromatography was performed using silica gel (60–120, 100–200, 230−400 mesh size; Merck). ^1^H NMR spectra were recorded (Brucker Avance, Zurich, Switzerland) at 400 and 500 MHz, and ^13^C NMR was performed at 100 and 125 MHz in CDCl_3,_ DMSO d_6_ and CD_3_OD. The chemical shift values are reported in δ (ppm) units, and coupling constants values are in hertz. Tetra methyl silane (TMS) was used as an internal standard. The following abbreviations were used to explain multiplicities: s = singlet, d = doublet, t = triplet, q = quartet, m = multiplet, and br = broad.

### 4.4. Extraction, Fractionation and Isolation of Standard Compounds

The powdered rhizome (2 kg) was subjected to maceration using ethanol (70%) as a solvent at room temperature for 12 h. The process was repeated thrice with the same solvent until a clear solution was obtained. The extract so obtained was filtered (Whatman filter paper No.2), and then the solvent was dried using a vacuum evaporator while keeping the temperature below 40 °C. The filtrate was concentrated and combined to afford a residue of 382.00 g (19.1% *w/w*). The dried ethanolic extract was preserved in an airtight container at a temperature of –4°C.

The extract was suspended in water and then subjected to fractionation using solvents of varying polarity, i.e., hexane, chloroform and *n*-butanol. The different fractions so obtained were dried on a rotavapor under reduced pressure. Further, the chloroform and *n*-butanol fractions of the rhizome extract were subjected to column chromatography for the isolation of bioactive compounds.

The chloroform fraction (74.3 g) was subjected to column chromatography through 60–120 mesh size silica gel. The column was eluted using a gradient mixture of petroleum ether and ethyl acetate in ratios varying from 100:0 to 20:80, from which 24 fractions were obtained. The fractions were collected up to a volume of 250 mL. On the basis of similarities in TLC, these fractions were combined, resulting in twelve subfractions. Subfraction no. 8, weighing 8.45 g, was further purified by using silica gel with a mesh size of 100–200. A solvent system comprising a mixture of petroleum ether–ethyl acetate in ratios ranging from 95:5 to 15:85 was used for the elution of the column. A volume of 150 mL was collected for each fraction to afford ten smaller subfractions (Fr. 10a–10j). Fraction 10e (1.430 g),upon final purification with silica gel with 200–400 mesh size in a solvent system of ethyl acetate and petroleum ether mixture (95:5 to 40:60%), gave fifteen smaller fractions (15a–15o), each having a volume of 50 mL. Precipitates were formed in fraction number six when kept overnight. These precipitates were further purified by washing with HPLC-grade hexane, yielding 258 mg of the pure compound, which was coded as compound 1. Compound 1 was analyzed by TLC (petroleum ether–EtOAc, 85:15) as a purple spot by placing the dried TLC plate in a UV chamber and visualized as a dark purple spot by staining the plate with anisaldehyde reagent and heating it at 105 °C. Further, fraction 10f (334 mg) was analyzed by TLC, which showed the presence of a prominent compound. The fraction was chromatographed over 200–400 mesh size silica gel using a solvent system of a petroleum ether–ethyl acetate gradient mixture with ratios ranging from 90:10 to 40:60, giving a pure compound that precipitated out in smaller fractions. The compound was further purified by washing with HPLC-grade hexane, yielding 142 mg of the pure compound, coded as compound 2. Compound 2 was analyzed by TLC (petroleum ether–EtOAc, 85:15) as a dark-green-colored spot by placing the TLC plate in a UV chamber and staining it with anisaldehyde reagent after heating at 105 °C. The *n*-butanol fraction (10 g) was subjected to a DIANION HP 20 resin column using a gradient solvent system consisting of 100% H_2_O to 100% MeOH, giving ten fractions, SC 1–10. Fraction SC-4 (0.85 g) was further purified using RP-HPLC, where the chromatographic conditions applied for purification were as follows: Column ACE 5 C18, flow rate 2.5 mL/min, column temperature 25 °C, and solvent system in the gradient elution method composed of0–6 min 5% ACN in H_2_O, 6–12 min 10% ACN in H_2_O, 12–18 min 70% ACN in H_2_O, 18–22 min, 10% ACN in H_2_O, 22–24 min and 5% ACN in H_2_O. The target peaks were collected at retention times of 13.3–14.9 min and 8.0–10.3 min. The two collected peaks were dried in a rotatory evaporator and were coded as compound 3 (25.0 mg) and compound 4 (26.9 mg).The structures of all isolated compounds (1–4_) were identified by 1D and 2D NMR spectral techniques.

### 4.5. HPLC Instrumentation and Chromatographic Conditions

HPLC analysis was performed on a Thermo fischer HPLC system (Chromeleon software version 7.2.9, Karlsruhe, Germany). The HPLC instrument was composed of a diode array detector (DAD), an autosampler, a degassing unit, a quaternary pump and a column oven. The detection was found to be optimal at 220 nm. Chromatographic separation was carried out using a lithosphere RP 18 end-capped column (250 mm × 4 mm, 5 µm) while keeping the temperature at 35 °C. The mobile phase was optimized with0.1% formic acid in water as solvent A and methanol as solvent B. The injection volume of the crude extract, fractions and standard marker compounds was kept at 10 μL. The solvent flow was optimized at 0.8 mL/min so as to obtain good separation between different peaks in the crude extract and different fractions. The temperature of the sampler was kept at 10 °C.

### 4.6. Preparation of Standard Solution and the Calibration Curve

Standard stock solutions of 1 mg/mL were prepared with HPLC-grade methanol and kept at 4 °C. All standards were filtered through a 0.22μm syringe filter before HPLC analysis. The standard samples were diluted with HPLC-grade methanol to provide a series of different concentration ranges for validation and calibration. The calibration curve was obtained by using ten different concentrations of each compound, which were as follows: 0.98, 1.98, 3.91, 7.8, 15.63, 31.2, 62.5, 125.0, 250.0 and 500.0 µg/mL for each compound.

### 4.7. Cell Culture

The cancer cell lines A549, HCT-116, MDA-MB 231 and PC-3 used for cytotoxicity evaluation were obtained from the National Centre for Cell Science, (Pune, India). These cell lines were cultured in complete growth medium (RPMI-1640 and DMEM) supplemented with 10% Fetal Bovine Serum, 100 µg/mL Streptomycin and 100 units per ml penicillin (Gibco, Thermo Fisher, Waltham, MA, USA), and the temperature was maintained at 37 °C with 5% carbon dioxide and 98% relative humidity.

### 4.8. Cytotoxicity Assay

The Sulforhodamine B (SRB) test was performed for the determination of the anticancer activity of the ethanolic crude extract and different fractions. A cell suspension with a cell density of 7500–15,000 cells/100µL was seeded in 96-well plates with flat bottoms for 24 h. After 24 h of incubation, the cells were treated with 3.125, 6.25, 12.5, 25, 50 and 100 µg/mL concentrations of the test materials for 48 h. Paclitaxel was used as a standard drug. After 48 h of incubation, cells were fixed using ice-cold 1% TCA for 1hr while keeping the temperature at 4 °C. The plates were washed thrice using distilled water and then left to dry in the air. After drying, 0.4% Sulforhodamine B (SRB) solution was added to each well and left for at least 30 to 40 min at room temperature. Finally, the SRB solution was removed from the plates by washing with 1% *v/v* acetic acid. The OD reading was taken at 540 nm using a Microplate reader (Thermo Scientific, Waltham, MA, USA) after dissolving the dye in 10 mM Tris buffer (pH-10.4) in each well. IC50 was calculated by plotting OD versus concentration using Graph PAD Prism (Version 6) (Boston, MA, USA).

## 5. Conclusions

In this study, four marker compounds, i.e., dehydrocostus lactone, costunolide, syringin and 5-hydroxymethyl-2-furaldehyde, were isolated from different fractions of the crude extract of *A. costus*. The simultaneous quantification of these four marker compounds was carried out on the crude extract and fractions by developing a suitable HPLC-PDA method. The method was validated by determining the correlation coefficient (r^2^), specificity, limit of detection, limit of quantification, intra- and interday precision, analyte recovery and stability of the standard compounds. Further, the anticancer activity results confirm that fractions rich in sesquiterpenes (dehydrocostus lactone and costunolide) with an exomethylene functionality possess significant anticancer potential. Further, the above findings will augment the phytochemical investigation of plants belonging to the Asteraceae family for the isolation of novel compounds and the preparation of their synthetic analogs for the treatment of cancer. The identified markers (1–4) and HPLC-PDA method can be used for the quality assessment of crude drugs and several Ayurvedic formulations in which this plant is used as a main ingredient.

## Figures and Tables

**Figure 1 molecules-28-04815-f001:**
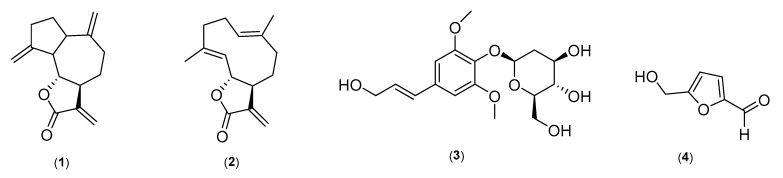
The isolated bioactive marker compounds from different fractions of *A. costus.* (**1**) Dehydrocostus lactone, (**2**) costunolide, (**3**) syringin and (**4**) 5-hydroxymethyl-2-furaldehyde.

**Figure 2 molecules-28-04815-f002:**
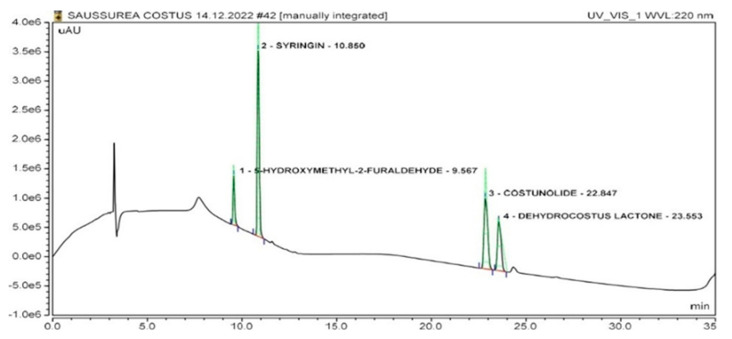
High-performance liquid chromatography (HPLC) chromatogram observed at 220 nm of a standard compound mixture from *A. costus* displaying the separation of isolated marker compounds (**1**) 5-hydroxymethyl-2-furaldehyde, t_R_ 9.56, (**2**) syringin, t_R_ 10.85, (**3**) costunolide, t_R_ 22.84, and (**4**) dehydrocostus lactone, t_R_ 23.55, and, obtained by the developed HPLC method.

**Figure 3 molecules-28-04815-f003:**
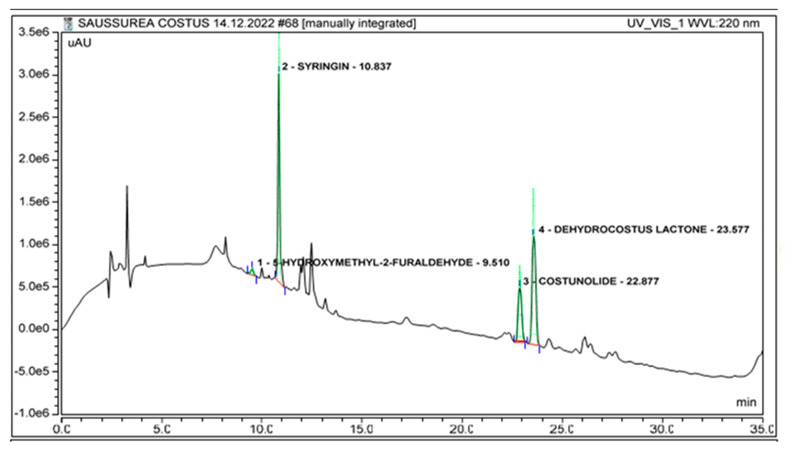
High-performance liquid chromatography chromatogram at 220 nm of crude extract of *A. costus* showing identified peaks of isolated marker compounds used as standard compounds for quantification in extract and fractions using the developed HPLC method.

**Figure 4 molecules-28-04815-f004:**
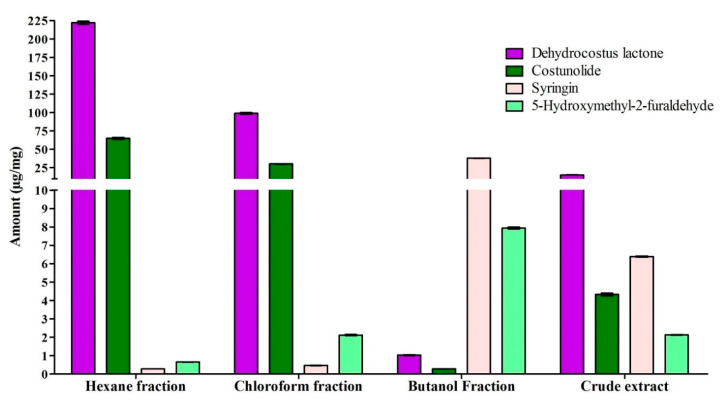
The relative contents of bioactive marker compounds dehydrocostus lactone, costunolide, syringin and 5-hydroxymethyl-2-furaldehyde in crude ethanolic extract and hexane, chloroform, and butanol fractions.

**Figure 5 molecules-28-04815-f005:**
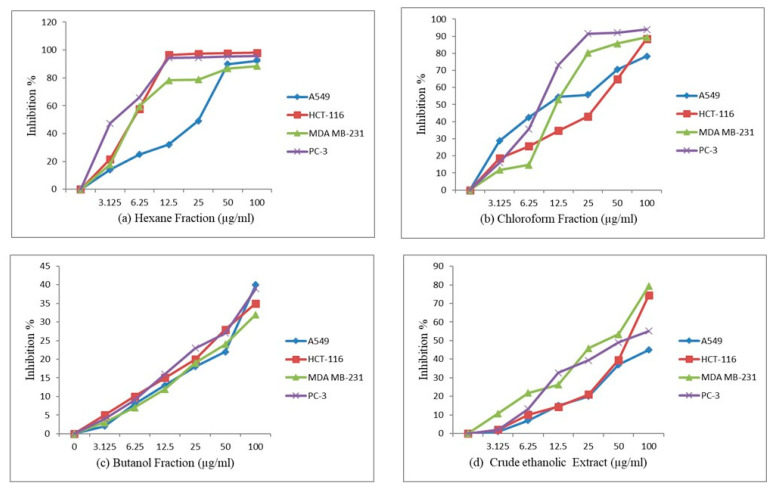
Cytotoxicity activity (% growth inhibition) for (**a**) hexane, (**b**) chloroform and (**c**) butanol fractions and (**d**) crude ethanolic extract at different concentrations of 3.125, 6.25, 12.5, 25, 50 and 100 µg/mL.

**Table 1 molecules-28-04815-t001:** Calibration curve parameters for standard compounds in the developed HPLC method used for analysis.

No.	Compound	Retention Time (Minutes)	r^2^	LOD (µg/mL)	LOQ (µg/mL)
1.	Dehydrocostus lactone	23.55 ± 0.04	0.9998	0.22	0.66
2.	Costunolide	22.84 ± 0.04	0.9999	0.12	0.36
3.	Syringin	10.85 ± 0.04	0.9938	0.11	0.33
4.	5-Hydroxymethyl-2-furaldehyde	9.56 ± 0.04	0.9998	0.30	0.90

**Table 2 molecules-28-04815-t002:** System suitability parameter evaluation of four marker compounds.

No.	Compound	Asymmetry	S/N Ratio	No. of Theoretical Plates
1.	Dehydrocostus lactone	1.30	52	56922
2.	Costunolide	1.56	44.3	49607
3.	Syringin	1.39	46	54850
4.	5-Hydroxymethyl-2-furaldehyde	1.41	62.2	60769

**Table 3 molecules-28-04815-t003:** Intra- and interday precision values of marker compounds for the developed HPLC method.

Precision Intraday					Interday	
No	Compound	Actual Conc. of Analyte (µg/mL)	Observed Conc. of Analyte (µg/mL)	RSD (%)	Observed Conc. of Analyte (µg/mL)	RSD (%)
1.	Dehydrocostus lactone	10.00	9.31	1.07	9.35	0.05
		100.00	100.15	0.05	100.17	0.02
		250.00	247.16	0.06	248.36	0.12
2.	Costunolide	5.0	5.15	0.78	5.09	0.30
		50.00	51.15	0.09	52.10	0.09
		125.00	125.78	0.12	124.52	0.08
3.	Syringin	10.0	10.60	1.70	10.65	0.20
		100.00	101.50	0.33	100.57	0.28
		250.00	252.81	0.23	241.68	0.34
4.	5-Hydroxymethyl-2-furaldehyde	5.00	5.64	1.51	5.19	1.85
		50.00	50.25	0.19	50.16	0.05
		125.00	125.24	0.18	124.91	0.22

**Table 4 molecules-28-04815-t004:** Recovery data of standard compounds dehydrocostus lactone, costunolide, syringin and 5-hydroxymethyl-2-furaldehyde using the developed HPLC method.

No	Compound	Original Conc. (µg/mL)	Spike Conc. (µg/mL)	Observed Conc. (µg/mL)	%Recovery ± SD	RSD (%)
1.	Dehydrocostus lactone	5.0	0.50	5.55	110.0 ± 2.00	1.81
			0.85	5.92	108.96 ± 1.78	1.64
			1.13	6.35	104.22 ± 1.76	1.69
2.	Costunolide	4.50	0.85	5.33	97.62 ± 1.16	1.94
			1.50	6.07	105.02 ± 1.46	1.39
			2.0	6.54	102.38 ± 1.08	1.06
3.	Syringin	3.0	0.45	3.45	101.77 ± 0.66	0.65
			0.90	3.92	103.03 ± 0.75	0.73
			1.35	4.41	102.43 ± 1.07	1.05
4.	5-Hydroxymethyl-2-furaldehyde	6.0	0.50	6.55	110.20 ± 2.00	1.81
			1.25	7.21	97.52 ± 0.36	0.37
			1.75	7.74	101.14 ± 0.431	0.42

**Table 5 molecules-28-04815-t005:** Stabilities of standard compounds in their solutions under storage conditions of room temperature (RT) and 4 °C observed for time periods of 0, 6, 12, 24 and 48 h.

Peak Area (AU ± SD)								
No	Compound	Temp	0 h	6 h	12 h	24 h	48 h	RSD%
1.	Dehydrocostus lactone	R. T	353,737.6 ± 219.4	351,436.3 ± 389.7	349,150.6 ± 277.3	341,010.6 ± 245.4	339,356.3 ± 251.2	1.84
		4 °C	353,461.3 ± 186.2	351,467.3 ± 188.4	349,076.6 ± 190.0	340,680 ± 95	338,091 ± 123.1	1.95
2.	Costunolide	R. T	518,111.3 ± 230.9	517,746.6 ± 205.9	516,795.3 ± 179.1	504,090 ± 278.74	489,646 ± 327.04	2.22
		4 °C	518,090.3 ± 171.7	517,310.6 ± 174.18	516,308.3 ± 175.5	503,310 ± 295.46	488,431.6 ± 267.5	2.52
3.	Syringin	R. T	322,582 ± 182.4	321,731 ± 295.9	321,069 ± 177.6	320,186.6 ± 255.9	305,507.3 ± 335.11	2.24
		4 °C	322,186 ± 212.03	321,571.3 ± 250.63	320,767 ± 258.2	319,787 ± 338.04	294,418.6 ± 267.23	3.78
4.	5-Hydroxymethyl-2-furaldehyde	R. T	150,886.6 ± 245.8	149,345.6 ± 200	147,433.3 ± 591.6	143,536.6 ± 228.10	141,447.3 ± 389	2.69
		4 °C	150,116.6 ± 309.8	148,952 ± 245.3	145,400.6 ± 443.5	142,965 ± 258.98	140,599.3 ± 379.6	2.73

**Table 6 molecules-28-04815-t006:** In vitro cytotoxicity of the crude extract and hexane, chloroform and butanol fractions.

		IC50 (µg/mL) ± SD			
No	Sample Code	A549 (Lung Cancer Cell Line)	HCT-116 (Colon Cancer Cell Line)	MDA-MB 231 (Human Breast Cancer Cell Line)	PC-3 (Prostate Cancer Cell Line)
1	Ethanolic extract	>100	64.05 ± 2.26	35.766 ± 2.20	52.998 ± 2.18
2	Chloroform fraction	11.875 ± 0.84	11.53 ± 0.42	12.179 ± 0.32	7.527 ± 0.18
3	Hexane fraction	20.904 ± 0.75	4.717 ± 0.16	5.353 ± 0.13	3.37 ± 0.14
4	Butanol fraction	>100	>100	>100	>100

## Data Availability

The data presented in this study are available on request from the corresponding author.

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
