# Peer review of "HPLC-PDA Method for Quantification of Bioactive Compounds in Crude Extract and Fractions of Aucklandia costus Falc. and Cytotoxicity Studies against Cancer Cells"

_molecules, 2023, doi:10.3390/molecules28124815_

Round 1

Reviewer 1 Report

Abstract: The abstract is long and describes each part of the manuscript. I recommend dividing it into Background, Aims, methodology, and crucial conclusions and highlighting the essential ones. The abstract need to rewrite and adjusted to a maximum of 200 words.

According to WFO Plant List, the correct name of the plant species of this study is Aucklandia costus Falc., the authors clarify that Saussurea costus is a synonym. Thus, Why is it the idea to use a synonym and not the correct name? In this case, I suppose a “common” name or different names exists for this plant. In this case, I suggest adding and clarifying the accepted name (Aucklandia costus Falc.). I do not find the relevance of citing a synonymous name; if there is one, the authors should clarify this situation.

Introduction

The first paragraph of the introduction is repetitive, and the idea has been reported and raised in different manuscripts. It is known that secondary metabolites from plants are a valuable source that could become a drug and insert in the pharmacy. Furthermore, the authors did not cite the gold reference of this topic from Newman & Cragg ( DOI: 10.1021/acs.jnatprod.9b01285). I do not see the novelty in this paragraph, and I suggest they rewrite the idea using novel information and the Newman & Cragg reference.

The authors mentioned that their work aimed to develop an analytical method using HPLC. But, What type of detector do they use? And like a second aim, their evaluated anticancer potential. Which type of cancer cell? Which type of crude extract or metabolites? The authors must clarify this idea and cover more about these topics in the introduction.

In general, the introduction could be improved, and it doesn't represent the state of the art their work tries to present. In this sense, the title is ambiguous, and this need to add what type of HPLC was used (MS, MS/MS, UV, UV-VIS, UV-DAD, FLD, etc), quantification of what? Anticancer potential of which type of cells? The title needs to be clarified.

Results

Figures 2 and 3 are not cited in the text.

Figure 4 and Table 2 present the same results. In this case, I recommend only presenting figure 4.

The authors present a correct validation of the analytical methodology, which is successful with international guides.

In Anticancer activity. Why don't the authors evaluate all the fractions? Furthermore, it evaluated the Ethanolic fraction, which was not validated previously. Please, clarify this information.

Discussion

The discussion is terrible and could be improved. The authors do not cite any work. Only describe their results again.

First, the authors in the introduction present two aims, one, the analytical method, and second anti-cancer activity. Thus, they need to discuss their results compared with the works in the literature and the advantage of the analytical method. In this case, the authors use a traditional method, HPLC-UV-VIS. Why select this technique? Second, the part of the anti-cancer activity has much information to discuss. The authors have yet to compare it with other drugs o natural products. Please, review recent literature and compare IC50 and the cancer cell line. These comments are mandatory to improve if authors pretend to publish their work.

The work and idea are interesting, and the first part of the work (analytical method) is correct and present enough evidence of their results. However, the second part needs restructuring and improving the discussion. Remember that it is a journal and not a simple report. For that, at this moment, I recommend rejecting this manuscript.

Reviewer 2 Report

The manuscript "HPLC Quantification and Evaluation of Anticancer Potential of Bioactive fractions and crude extract of medicinal plant Saussurea costus (Falc.) Lipsch." reports the isolation and quantification of main compounds found in crude extract S. costus and evaluation of the anticancer activity and the evaluation of their anticancer activity. The isolation, chemical characterization and the development of a robust and suitable method to analyze and quantify some main natural compounds in the extract of S. costus are widely described, providing a valid method to assess the quality of S. costus extracts. However, the biological effects of these substances are poorly characterized and the claimed anticancer potential is based on very preliminary data.  Moreover, figure 5 and table 7 report exactly the same data in a different form (table and histogram). It would be better reporting the log dose/response curve from which the IC50 values were calculated. Additionally, the description of the cytotossicity assay lacks some information.  Authors state "The cultures were incubated with 3.125 µg, 6.25 µg, 12.5 µg, 25 µg, 50 µg, 100 µg concentrations of test material in a complete growth medium (100 μL) after 24 h of incubation" but which were the effective concentrations used (3.125 µg in 100μL would mean 31.25 µg /ml?).

Abstract is too long. Information regarding the analytical methods description can be removed.

Round 2

Reviewer 1 Report

The authors attended to all the comments and improved and rewrote the introduction and discussion according to their aims. Right now, I would recommend publishing in the journal. 

The authors need to review the template and adapted the information to this. 

Reviewer 2 Report

Authors have adequately addressed  my comments  in the revised version of the manuscript.
